# Molecular Characterization of Tc964, A Novel Antigenic Protein from *Trypanosoma cruzi*

**DOI:** 10.3390/ijms21072432

**Published:** 2020-03-31

**Authors:** Elizabeth Ruiz-Márvez, César Augusto Ramírez, Eliana Rocío Rodríguez, Magda Mellisa Flórez, Gabriela Delgado, Fanny Guzmán, Paulino Gómez-Puertas, José María Requena, Concepción J. Puerta

**Affiliations:** 1Grupo de Investigación en Enfermedades Infecciosas, Departamento de Microbiología, Facultad de Ciencias, Pontificia Universidad Javeriana, Carrera 7 # 40- 62, Bogotá, Colombia; elizabeth.ruiz@javeriana.edu.co (E.R.-M.); Ramirez-Segura.Cesar@mayo.edu (C.A.R.); erodriguezg@javeriana.edu.co (E.R.R.); 2Grupo de Investigación en Inmunotoxicología, Departamento de Farmacia, Facultad de Ciencias, Universidad Nacional de Colombia, Carrera 30 # 45-01, Bogota; Colombia; mmflorezm@unal.edu.co (M.M.F.); lgdelgadom@unal.edu.co (G.D.); 3Núcleo de Biotecnología Curauma (NBC), Pontificia Universidad Católica de Valparaiso, Avenida Universidad 2373223, Curauma, Valparaiso-Chile; fanny.guzman@pucv.cl; 4Grupo de Modelado Molecular del Centro de Biología Molecular Severo Ochoa, Microbes in Health and Welfare Department, Universidad Autónoma de Madrid (CBMSO, CSIC-UAM), 28049 Madrid, Spain; pagomez@cbm.csic.es; 5Grupo Regulación de la Expresión Génica en Leishmania del Centro de Biología Molecular Severo Ochoa, Molecular Biology Department, Universidad Autónoma de Madrid (CBMSO, CSIC-UAM), 28049 Madrid, Spain; jmrequena@cbm.csic.es

**Keywords:** chagas disease, leishmaniasis, *Leishmania major*, Trypanosoma cruzi, molecular modelling, serodiagnosis

## Abstract

The Tc964 protein was initially identified by its presence in the interactome associated with the LYT1 mRNAs, which code for a virulence factor of *Trypanosoma cruzi*. Tc964 is annotated in the *T. cruzi* genome as a hypothetical protein. According to phylogenetic analysis, the protein is conserved in the different genera of the Trypanosomatidae family; however, recognizable orthologues were not identified in other groups of organisms. Therefore, as a first step, an in-depth molecular characterization of the Tc946 protein was carried out. Based on structural predictions and molecular dynamics studies, the Tc964 protein would belong to a particular class of GTPases. Subcellular fractionation analysis indicated that Tc964 is a nucleocytoplasmic protein. Additionally, the protein was expressed as a recombinant protein in order to analyze its antigenicity with sera from Chagas disease (CD) patients. Tc964 was found to be antigenic, and B-cell epitopes were mapped by the use of synthetic peptides. In parallel, the *Leishmania major* homologue (Lm964) was also expressed as recombinant protein and used for a preliminary evaluation of antigen cross-reactivity in CD patients. Interestingly, Tc964 was recognized by sera from Chronic CD (CCD) patients at different stages of disease severity, but no reactivity against this protein was observed when sera from Colombian patients with cutaneous leishmaniasis were analyzed. Therefore, Tc964 would be adequate for CD diagnosis in areas where both infections (CD and leishmaniasis) coexist, even though additional assays using larger collections of sera are needed in order to confirm its usefulness for differential serodiagnosis.

## 1. Introduction

Chagas disease (CD) or American Trypanosomiasis and its etiological agent (*Trypanosoma cruzi)* were first described by Dr. Carlos Chagas in 1909 [1,2]. A century later, this pathology remains a public health problem in Latin America, where it is associated with poverty conditions [3,4] and belongs to the so-called neglected tropical diseases [5,6]. Currently, 8 million people in the world are infected with *T. cruzi* [5]. Southern Cone countries such as Brazil, Argentina, and Uruguay contribute with about 62.4% of cases of infection, and Colombia in 2015 contributed with 437,960 cases out of a total of 958,453 reported in the Andean region [7]. In recent decades, due to recurring migration patterns from Latin America, the disease has been spread to different areas around the world such as Europe, Japan, Australia, Canada and the United States; in the latter country, more than 300,000 infected people are living there [8,9].

Chagas disease evolves through two clinical stages, recognized as the acute and chronic phases. The former corresponds to the initial stage of the infection, and its duration lasts between one to two months [10,11]. In this phase, trypomastigotes of *T. cruzi* can often be observed by light microscopy in the blood of patients; however, due to the absence of symptoms only 5% of them are diagnosed [3]. Most of the infected persons enter the asymptomatic or indeterminate chronic phase, in which the parasite is barely observed, and no clinical symptoms occur. Nevertheless, 10 to 30 years after the initial infection, about 30 to 40% of the patients progress to the symptomatic or determinate chronic phase, developing cardiac arrhythmias, progressive heart failure or digestive alterations as megaesophagus and megacolon. These alterations are linked to the tissue tropism of the parasite; during the chronic stages of the disease, *T. cruzi* mainly persists in the cardiac and mesenteric muscles. Although the parasitaemia in chronic CD (CCD) patients is very low or undetectable, the presence of high levels of antibodies against *T. cruzi* antigens [12] serves for diagnosing CCD [13,14].

The World Health Organization and Pan American Health Organization [5,6] recommend for patients with suspected of chronic infection by *T. cruzi* to perform the standard diagnosis. This consists of two serologic tests that use different principles such as indirect immunofluorescence, indirect hemagglutination or enzyme-linked immunosorbent assay (ELISA), which may differ in the type of antigens employed, e.g., parasite total extracts, recombinant proteins or synthetic peptides [15,16,17]. For this purpose, different proteins and antigens of *T. cruzi* have been characterized and evaluated for serodiagnosis [17]. However, many of them show cross-reactivity with sera from patients infected with other trypanosomatids, particularly *Leishmania* spp; these parasites share habitats and hosts, and serodiagnosis of CD and cutaneous, mucocutaneous and visceral leishmaniasis may give mistaken results. Therefore, the characterization of new non-cross-reactive *T. cruzi* antigens is pivotal to generate a reliable CD serodiagnosis [15,18].

In this work, we report the characterization of a novel protein from *T. cruzi*, named Tc964, which was initially identified by mass spectrometry due to its binding to the 5′ untranslated region of the LYT1 mRNAs [19] within a project aimed to study the regulatory mechanisms operating on *LYT1* gene expression. The expression product of LYT1 is a protein associated with the escape of trypomastigotes from the parasitophorous vacuole in the mammalian host. In addition, LYT1 is involved in the transition from the epimastigote to the metacyclic trypomastigote stage, usually occurring in the triatomine insect vectors [20,21,22]. The Tc964 protein of *T. cruzi* was identified by mass spectrometry in a study aimed at detecting proteins associated to the untranslated regions of the LYT1 mRNAs [23]. Currently, Tc964 is annotated in databases as a “hypothetical protein”; however, in this study, we demonstrated its presence in the nucleus and cytoplasm of the parasite. Moreover, based on structural data, this protein may be a member of the GTPase superfamily. Additionally, Tc964 was specifically recognized by sera from CCD patients and, interestingly, the protein did not show cross-reactivity with sera from Colombian patients with cutaneous leishmaniasis. Therefore, the Tc964 protein (or particular antigenic peptides) may be considered as a potential marker for serodiagnosis of CD, warranting additional studies in the future.

## 2. Results

### 2.1. Tc964 Is Conserved in Different T. cruzi Discrete Typing Units (DTU) and Is Also Present in Other Kinetoplastids

Tc964 is encoded by the *TcCLB.511467.70* gene, according to the genome annotation for the CL Brener Esmeraldo-like strain available at the GeneDB database [24]. The gene is located at chromosome 38 (Figure 1A). Furthermore, we identified the gene in the genomes for other *T. cruzi* DTU, and the sequence analysis showed a remarkable sequence conservation (Appendix A). The accession numbers obtained for them from the TritrypDB [25] and the GenBank databases [26] are registered in Appendix A [27,28,29,30,31,32]. Also, we searched for homologous proteins in other kinetoplastids; Tc964 orthologues were identified in all the kinetoplastids analyzed (Figure 1B). Orthologues were identified in species of the *Trypanosoma* genus, *T. rangeli* (79% of sequence identity) and *T. brucei* (50%), in the *Leishmania* genus (48%), in monoxenous trypanosomatids (47%) and in the free-living *Bodo saltans* (32%). The phylogenetic relationships inferred from the tree based on Tc964-orthologues fit well with current taxonomy of kinetoplastids [33]. However, by BLAST searches at NCBI, no clear orthologues were found in organisms other than kinetoplastids.

### 2.2. Tc964 Protein May Be a Member of the GTPase Family

Due to the lack of a clear homology between the sequence of Tc964 and other proteins of known function, a combined sequencing-structure compatibility approach was used to assign a possible function to the protein.

Using PSI-BLAST [35] and Robetta programs [36], a putative remote homology, supported by a 18% global sequence identity, was found between Tc964 and an archaeal GTPase protein, named SRP54. This protein is part of a ribonucleoprotein complex in which SRP54 forms a heterodimer with the protein FtsY. The crystallographic structure of this heterodimer was solved recently (PDB code: 5L3S) [37]. Alternatively, using the PHYRE2 program, a 22% identity was found between Tc964 and the glycolytic enzyme enolase from *Escherichia coli* (PDB code: 2FYM) [38]. According to the Robetta, a 19.9% sequence identity exists between Tc964 and *Plasmodium falciparum* Lys-tRNA ligase (PDB code: 5ZH3) [39].

Using 3D structures of SPR54, *E. coli* enolase, and *P. falciparum* tRNA ligase as templates, three structural models of the Tc964 protein were obtained by modeling tools (see the Methods section for specific details). Afterwards, each predicted Tc964 structure was embedded in a box of virtual solvent and subjected to a 50 nanoseconds simulation of free molecular dynamics, using the AMBER program [40], to test its structural stability. During this simulation, the Root Mean Square Deviation was continuously calculated for the three models; this parameter indicates how much the modelled structure deviates from the initial arrangement during the molecular dynamics trajectory. Thus, Root Mean Square Deviation offers an estimate of the stability of the model and, therefore, its reliability: the higher its value and/or the greater the oscillations of this value, the lower the confidence in the model. Only the model based on the SRP54 template [37] remains stable along the whole molecular dynamics simulation (Figure 2A) and, therefore, this was the structure chosen to build a probable model representing the spatial structure of the Tc964 protein (Figure 2A). A similar procedure was followed using the sequence of the *L. major* (Friedlin strain) orthologue, named Lm964 (GeneDB code: *LmjF.35.0160*); again, the more stable conformation was obtained when SRP54 was used as template. Indeed, the characteristic N, M and G domains from SRP54 were also predicted in both Tc964 and Lm964 proteins. All SRP54 proteins, from archaea as *Saccharolobus solfataricus* (SRP54 template) or bacteria as *Thermus aquaticus* to *Homo sapiens*, include the presence of a flexible linker region, containing the highly conserved motif “LGMGD”. This motif allows direct contact between the G and M domains in this class of GTPases [41]. In the Tc964 protein, this linker region seems to be conserved at amino acids positions 222 to 224 (Figure 2C). The first three amino acids that are part of the LGMGD binding motif are located in a loop which connects G and M domains. Interestingly, the same three amino acids are present in the linker region of the Tc964 and Lm964 proteins (Figure 2C). This linker motif is conserved in *T. cruzi*, *T. rangeli*, *T. grayi* and in all the species of the *Leishmania* genus (Appendix A).

In summary, based on the modelling results, Tc964 and Lm964 proteins may represent distant relatives to the archaeal SRP54 GTPase. Some specific structural differences were observed between the SPR54 and Tc964 protein such as: (i) The N domain (amino acids 1 to 65) of the 964 protein of *T. cruzi* and *L. major* lacks the alpha IV helix present in the SPR54 template (orange helix in Figure 2B); (ii) The structure of beta sheets in the G domain (amino acids 66 to 221), which allows GTP binding, is not well defined in the 964 protein of *T. cruzi* and *L. major* (Figure 2) and (iii) The M domain present in the SRP54 template, which provides a binding site for the signal peptide and is connected to the G domain, is apparently incomplete in the 964 protein (amino acids 222 to end) (Appendix A) [42].

### 2.3. Tc964 is a Nucleocytoplasmic Protein

The subcellular localization of Tc964 was investigated by analyzing the presence of the protein in nuclear and cytoplasmic fractions from epimastigotes and trypomastigotes of *T. cruzi*. The purity of the subcellular fractions was evaluated by using antibodies against histone H3 (nuclear) and Heat-Shock Protein 70 (essentially cytoplasmic). The Western blots showed a 35 kilodaltons (kDa) protein band in both subcellular fractions (Figure 3A,B. First lane), which is consistent with the theoretical molecular weight (311 amino acids and 34.7 kDa). Upper molecular species, around 45 kDa, were observed, mainly in the trypomastigote forms (Figure 3A, B), suggesting a possible post-translational modification of the protein along the parasite development.

In support of the nuclear location of Tc964, we predicted the presence of two nuclear location motifs in the Tc964 protein, i.e., a non-classical Nuclear Localization Signal composed by the PRVRY and NPYTTRP motifs and a Nuclear Export Signal rich in Leucine (L) [45,46]. Regarding the NLS sequence, in the PRVFY motif, the Proline (P), the first Arginine (R), Valine (V), and Tyrosine (Y) residues, are identical (amino acids 105–109) (Appendix A). The second motif, NPYTTRP (amino acids 235–241), conserves the Asparagine (N) and two (P) residues (Appendix A). Finally, the Nuclear Export Signal rich in (L) matches the consensus sequence φ-X2-3-φ-X2-3-φ-x- φ (amino acids 184–207), where φ can be L, I, V, F or M residues and X, any amino acid (Appendix A) [46].

### 2.4. Tc964 is Recognized by Sera from Chronic Chagas Disease Patients

With the purpose of testing the antigenicity of Tc964, firstly, sera from patients with CCD at different stages of the disease as well as sera from patients with leishmaniasis were used. The reactivity of the sera was determined by ELISA against soluble antigens obtained from lysates of *T. cruzi* epimastigotes and *L. major* promastigotes, respectively. Afterwards, sera with a Reactivity Index (RI) greater than 1, were used to assess their reactivity against the recombinant proteins from *T. cruzi* (rTc964) and *L. major* (rLm964).

The soluble antigens of *T. cruzi* were recognized by sera of patients with CCD in all the stages of severity evaluated, finding differences in the reactivity between the symptomatic groups C and D (*p*= 0.02), obtaining higher RI values in sera of the C group (Figure 4A). Meanwhile, 7 out of 12 sera from Active Cutaneous Leishmaniasis (ACL) patients and 4 out of 11 sera from Cured Cutaneous Leishmaniasis (CCL) patients reacted against the soluble antigens of *L. major*, and no differences in reactivity were observed between the ACL and CCL groups (Figure 4B).

Reactivity against the rTc964 protein was observed with CCD sera from both asymptomatic (A and B groups) and symptomatic patients (C and D groups). The sera from the B group had lower reactivity than the A (*p* = 0.0427), C (*p* = 0.0004), and D (*p* ≤ 0.0001) groups (Figure 4C). On the other hand, 5 out of 7 sera of patients with ACL and the four CCL sera reacted against the rLm964 protein and did not show significant differences among them (Figure 4D).

### 2.5. Tc964 Protein is Not Recognized by Sera from Leishmaniasis Patients

In order to analyze the specificity of the recombinant proteins, a cross-reactivity test was performed using the five most reactive sera from each group (four CCD stages and two leishmaniasis forms) and both recombinant proteins. Remarkably, while cross-reactivity between the rLm964 protein and sera from CCD patients was observed in the four groups of patients evaluated (Figure 5A), the rTc964 protein was not recognized by any of the sera from patients with leishmaniasis (Figure 5B). This result was verified through a Western blot assay using one of the most reactive sera from each patients group (Figure 5C).

### 2.6. Two Peptides Derived from Tc964 Protein are Recognized by Sera from Chronic CD Patients

With the aim of determining possible linear epitopes in the Tc964 protein, a B epitope prediction was conducted using several bioinformatics tools. Two potential epitopes, named TcNV and TcKP were identified, and chemically synthetized. These synthetic peptides were used in ELISA tests; both peptides were recognized by the nine most reactive sera of CCD patients, but they did not react with leishmaniasis sera (Figure 6). Interestingly, for the TcNV peptide, statistically significant differences were obtained between sera from asymptomatic and symptomatic patients (A and C (*p* = 0.00003), A and D (*p* = 0.0184), B and C (*p* ≤ 0.0001), B and D (*p* = 0.0003)) (Figure 6A). For the TcKP peptide, only the B and C groups showed significant differences (*p* = 0.0289) (Figure 6B).

## 3. Discussion

The characterization of proteins from pathogens of medical importance, as *T. cruzi,* is relevant because it provides information on different biological aspects of the parasite and allows assessing properties of these proteins to define them as targets for developing disease intervention strategies [47]. Based on this rationale, herein we undertook the molecular characterization of a novel *T. cruzi* protein, Tc964. Additionally, part of this work was aimed to analyze the antigenicity of the protein in Chagas disease patients.

In this study, it was possible to establish the presence of Tc964 in different taxa of the Kinetoplastid order, especially those of the Trypanosomatidae family. Significant sequence identity to Tc964 was not observed in organisms other than kinetoplastids. However, the Tc964 protein may be categorized as a member of the GTPase superfamily, based on structural predictions and molecular dynamics analysis using the SPR54 GTPase as template. This similarity is mainly patent in the presence, in both proteins, of N and G domains, crucial for the activity of GTPase as they are involved in the GTP hydrolysis [48,49]. SPR54 is part of a Signal Recognition Particle [49] and it is involved in the co-translational sorting of secretory and membrane proteins. In addition to its GTPase activity, the structure of this protein includes the presence of an RNA binding site. Unlike the SRP54 of mammals, bacteria and archaea, the Tc964 protein does not possess a canonical RNA binding site; remarkably, this binding motif is also absent in the SRP54 protein of chloroplasts [37,50]. Nonetheless, this does not rule out its capacity to bind RNA molecules as this protein was isolated by its association with a mRNA sequence (even though this binding may be the result of either a direct RNA or a protein mediated-RNA interaction). A question for the future will be to establish whether a functional role is shared between the proteins SPR54 and Tc964. Thus, it would be interesting to carry out an experimental analysis to find out if the Tc964 protein exhibits GTPase activity.

As an approach to elucidate the Tc964 functional aspects, we determined its subcellular location in epimastigotes and trypomastigotes of *T. cruzi*. The subcellular location influences the function of proteins while determining its microenvironment and the partners with which it interacts [51]. In this sense, it was possible to establish that Tc964 is a nucleocytoplasmic protein which possesses nuclear location signals that could address its shuttling to the nucleus. Remarkably, this pattern of subcellular location has been previously reported for GTPases of the Ras superfamily, such as Rab or Ran which are important adjuvants in the transport of cytoplasmic proteins to the nucleus, becoming relevant shuttling proteins for the correct orchestration of fundamental processes at the cellular level [52]. Consisting with its potential GTPase activity, the Western blot assays revealed a signal of higher molecular weight (45 kDa) which could correspond to a post-translational modification due to the binding of a ubiquitin molecule. In GTPases, ubiquitination is a common post-translational modification that allows these proteins to fulfill their function of maintaining the GTP cycle [53] and modulates the location of proteins as well as protein–protein interactions having direct effects on expression levels [54,55]. The study of the Tc964 interactome strengthens the hypothesis of a possible in vivo interaction of Tc964 with a protein from the *T. cruzi* ubiquitin family [56].

In parasites of medical importance, as several species of *Leishmania*, a relationship between the sequence conservation and the diagnosis potential of some evolutionarily conserved proteins has been established [16]. In this context, we explored the diagnosis potential of the conserved Tc964 protein. As expected, Tc964 was recognized by sera from CCD patients, confirming its antigenic properties, as previously reported for other kinetoplastid conserved proteins, such as the Kinetoplastid membrane protein-11 (KMP-11) [57] and TolT (TolA-like protein) proteins from *T. cruzi* [47]. Interestingly, sera from the B group of patients, who in spite of being asymptomatic already present structural heart damage, showed the lowest reactivity (Figure 4C). In this manner, changes in the Tc964 protein sera pattern recognition could be constituted as a possible early biomarker of disease progression. However, complementary research must be carried out to estimate if this reactivity pattern is extended to a higher number of CCD patients.

On the other hand, a crucial factor in obtaining reliable diagnoses in tropical infections with a sympatric distribution such as CD and leishmaniasis [12,18,58] is to assess the absence of cross-reactivity, which may occur between *T. cruzi* and *Leishmania* antigens. In this way, Tc964 and its TcNV and TcKP peptides were not recognized by cutaneous leishmaniasis sera patients, positive for anti-leishmania antibodies. However, to corroborate these preliminary results, it is necessary to perform additional analyses with a larger number of samples including those from patients with all the different clinical forms of leishmaniasis and other infectious diseases such as malaria and tuberculosis [15].

On the contrary, when Lm964 was used as antigen, CCD patient sera recognized it. These results could be explained by the fact that the *T. cruzi* specific epitopes are located in the non-conserved regions of the proteins, contrary to what happens with the *Leishmania* epitopes which could map to the conserved ones. Altogether, these findings prove, as other studies reported, that antigenic peptides can be a useful tool for the diagnosis of CCD [59,60,61].

## 4. Materials and Methods

### 4.1. Description of the TcCLB.511467.70 Gene in the Genome of T. cruzi

We searched in the TritrypDB [24] and GenBank [25] databases the gene coding for the Tc964 protein of *T. cruzi*. Its location in the genome and its relationship with orthologous genes in other species of the Trypanosomatidae family and Kinetoplastid order were established.

### 4.2. Phylogenetic Analysis of Protein Tc964

The evolutionary history of the Tc964 protein was constructed with the MEGA software, (version 7, Penn State, Philadelphia, PA, USA) [34], using 25 homologous protein sequences from parasites of Kinetoplastid order and Trypanosomatidae family retrieved from the TrytripDB database. The homologous sequence of the *B. saltans* species was used as an outgroup [62]. These sequences were aligned with the MUSCLE program and the phylograms were obtained using the Maximum Likelihood method, with the Jones–Taylor–Thornton model and the Minimum Distance method.

### 4.3. Fractionation and ELISA Test

#### 4.3.1. Cultivation of Parasites

One million of epimastigote forms of *T. cruzi* MHOM/BR/00/Y (DTU II) [63] were grown in Liver Infusion Tryptose (LIT) medium supplemented with 10% bovine serum albumin (Eurobio, Les Ulis, France) at 26 °C; after 96 h of culture, 2 × 10^8^ epimastigotes were obtained. In the meantime, Green Monkey renal fibroblast-like cells (Vero cells; ATCC CCL-81, Manassas, VA, USA) were grown in Dulbecco’s modified eagle medium (DMEM) (Eurobio, Les Ulis, France) supplemented with 10% bovine serum albumin (Eurobio, Les Ulis, France), 2 mM L-glutamine, 100 U/mL penicillin, 100 mg/mL streptomycin, and 0.01 M hepes (Eurobio, Les Ulis, France) at 37 °C in a humid atmosphere with 5% CO_2_. After reaching semiconfluence, the Vero cells were incubated for 10 h with 3 × 10^7^
*T. cruzi* trypomastigotes MHOM/BR/00/Y (DTU II) [63], and the parasites were recovered at 96 h postinfection. Trypomastigotes with three cyclic passages in Vero cells were used in subcellular location assays. Similarly, 10^6^ promastigotes of *Leishmania major* MHOM/IL/81/Friedlin strain [64] were grown in Schneider’s insect medium (Sigma-Aldrich, St. Louis, MO, USA), supplemented with 20% bovine serum albumin (Eurobio, Les Ulis, France), and 1 μg/mL of 6-biopterin (Sigma-Aldrich, St. Louis, MO, USA) at 26 °C; after 72 h of culture 8 × 10^6^ promastigotes in logarithmic growth phase were obtained.

#### 4.3.2. Subcellular Location of the Tc964 Protein of *T. cruzi*

Epimastigotes and trypomastigotes of *T. cruzi* were collected and washed three times with 1 mL of Dulbecco’s phosphate buffered saline (PBS 1×) (Eurobio Les Ulis, France). Then, 1 × 10^8^ parasites were resuspended in 300 μL of freshly prepared buffer B1 1× (200 mM Tris-HCl pH 7.4, 40 mM KCl, 25 mM MgCl_2_), 0.5% nonidet P40 (NP-40), 1 mM PMSF, 5 mM β-mercaptoethanol and protease inhibitor cocktail complete mini EDTA-free (Roche, Mannheim, Germany). An amount of 30 µL was transferred and labeled as total protein lysate. The remaining sample was centrifuged at 974 *g* for 2 min at 4 °C. The supernatant obtained was stored as cytoplasmic fraction and the pellet was washed again with 1 mL of buffer B1, and centrifuged at 974 *g* for 2 min at 4 °C. Finally, the pellet was resuspended in 100 µL of 2× Laemmli buffer and an equal volume of the cytoplasmic fraction was resuspended in 4× Laemmli.

#### 4.3.3. Soluble Antigens from *T. cruzi* and *L. major*

For the ELISA tests, the soluble antigens from *T. cruzi* and *L. major* were obtained from the lysates in the logarithmic growth phase of epimastigotes of *T. cruzi* and promastigotes of *L. major,* which were washed with PBS 1×; then, 10 cycles of freezing and thawing were applied at −80 °C and 37 °C, respectively. The samples were centrifuged at 2400 *g* for 10 min to 4 °C and the supernatants were stored at −80 °C until use [65]. The concentration of the antigen was determined by the Bradford assay (Biorad, Hercules, CA, USA).

### 4.4. Western Blot Assay

The anti-Tc964 antibody was obtained from the service for the production of animal antibodies (Centro de Investigación y Desarrollo, CID-CSIC. Barcelona, Spain). Antibody purification was performed by applying the immunoaffinity chromatography method with Affigel 10 gel (Biorad, Hercules, CA, USA). The samples were electrophoresed on 12% SDS-PAGE gels and transferred to nitrocellulose membranes (Biorad, Hercules, CA, USA), at 200 milliamperes for 1 h on transfer buffer (25 mM tris base, 192 mM glycine, and 20% methanol). The blots were blocked with 5% skim milk solution for 1 h at room temperature and then the anti-Tc964 polyclonal antibody was diluted 1:200 in blocking solution and incubated for 1 h at room temperature. Detection was performed with peroxidase-conjugated anti-rabbit IgG 1:5000 and developed using super signal west pico chemiluminescent substrate (Thermo Scientific, Waltham, MA, USA). Nitrocellulose membranes were re-probed using Thermo Scientific^™^ Restore^™^ Western blot stripping buffer (Thermo Scientific, Waltham, MA, USA) and densitometric analyses were performed using Fiji software (version 2, Schindelin J, Madison, WI, USA) [66]. On the other hand, the recombinant proteins rTc964, rLm964 and Trigger Factor, were electrophoresed on 11% SDS-PAGE gels and the same transfer and Western blot protocols mentioned above were followed. The sera were diluted 1:50 and the detection was performed with anti-human IgG (lambda-chain specific) alkaline phosphatase A3150 (Sigma-Aldrich, St. Louis, MO, USA), diluted 1:5000. The Sigma fast^™^ bcip^®^/ nbt (5-bromo-4-chloro-3-indolyl phosphate/nitro blue tetrazolium, (Sigma-Aldrich, St. Louis, MO, USA) was used as the revealing solution.

### 4.5. Structural Prediction of the Tc964 protein in T. cruzi and L. major

Three parallel strategies were followed in the search for homologues that could be used as templates for the generation of structural models: PSI-BLAST [34] against protein sequences in the Protein Data Bank, the protein structure prediction servers Phyre2 (version 2, Kelley L, London, United Kingdom; http://www.sbg.bio.ic.ac.uk/phyre2) [67] and Robetta (The Baker lab, Seatle, WA, USA; http://new.robetta.org) [36]. Model coordinates were built using the SWISS-MODEL server (http://swissmodel.expasy.org) [68] and their structural quality were within the range of those accepted for homology-based structure (Anolea/Gromos/QMEAN4) [69]. Model geometries were optimized using energy minimization with Deep View (version 4.1, Swiss-PdbViewer, Lausanne, Switzerland; http://spdbv.vital-it.ch/) [70]. Modeled structures were subjected to molecular dynamics simulation using the AMBER14 molecular dynamics package (http://ambermd.org/; University of California-San Francisco, CA) [40]. Free MD simulations were performed using the PMEMD program of AMBER and the parm99 force field. After the equilibration phase, 50 ns of free molecular dynamics simulations were obtained for each structure. Root Mean Square Deviation of Cα trace was monitored along the trajectories, measuring it every 20 ps. MD trajectories were analyzed using VMD software (Visual Molecular Dynamics) [71]. Figures were generated using the Pymol Molecular Graphics System (Schrödinger, LLC, Portland, OR, USA) [43].

### 4.6. Cloning and Protein Expression

The gene *TCSYLVIO_000964* homologous to the reference gene *TcCLB.511467.70* was amplified by PCR from *T. cruzi* Tc058 genomic DNA (DTU I), using a set of specific primers which at their 5′ end contain 15 nucleotides shared with the pQE30 plasmid: Inf964F (5′-TCACCATCAC***GGATCC***ATGGCGTACCGGCGGAAA-3′) and Inf964R (5′- GCAGGTCGACCCGG***GGTACC***TCAGGAAGAAACAAATATGCT-3′). The restriction sites for the *Bam*HI and *Kpn*I enzymes used for cloning purposes, are shown in italics and bold in the DNA sequence. The PCR product was purified and cloned directly into the plasmid pQE30 (QIAGEN, Hilden, Germany) using the Infusion^®^ HD EcoDry^™^ cloning kit, according to the manufacturer’s instructions (Clontech, Mountain View, CA, USA). Afterwards, the recombinant clone was sent to sequencing service at Universidad de los Andes, Bogotá, Colombia. For protein expression, *E. coli* M15 cells were transformed and the bacterial culture was induced overnight at 20 °C with 0.5 mM isopropyl thiol-β-galactopyranoside (IPTG) (Sigma-Aldrich, St. Louis, MO, USA). The denatured recombinant protein was purified by affinity chromatography using a Ni^2+^–NTA-Agarose resin (QIAGEN, Hilden, Germany).

The *LmjF.35.0160* gene was amplified with the specific primers sets F35.0160-*Bam*HI-2-F (5′-GG***GGATCC***ATGTCCTATCGGCGCAAGGC-3′) and F35.0160-*Sal*I-2-R (5′- GG***GTCGAC***TA CACTGGTTGA AAAACCGTTTC-3′). The restriction sites for the enzymes used for cloning purposes are shown in italics and bold in the DNA sequence. For the PCR conditions, Thermo Scientific Phusion Hot Start II High-Fidelity DNA polymerase amplification protocol was used (Thermo Scientific, Waltham, MA, USA). The purified DNA fragment was ligated to the vector pNZY28A (nzytech genes and enzymes, Lisbon, Portugal) applying the ligation mixture from T4 DNA ligase for blunt ends (Thermo Scientific, Waltham, MA, USA). Once the cloning of the *LmjF.35.0160* gene was confirmed by sequencing (Parque Científico of Madrid [PCM], Spain), it was subcloned into the pET28a [+] [5.36 Kb] expression plasmid (Sigma-Aldrich, St. Louis, MO, USA), generating the pET28a-*LmjF.35.0160* recombinant plasmid. Afterward, *E. coli* Nico21 [DE3] was electroporated following the protocol from the Fermentation Service (Centro de Biología Molecular Severo Ochoa. Madrid, Spain). Soluble fractions of *E. coli* Nico21 [DE3] cultures induced for four hours at 37 °C with 0.5 mM IPTG (Sigma-Aldrich, St. Louis, MO, USA) were purified by gravity-flow chromatography with Ni^2+^-NTA agarose (QIAGEN, Hilden, Germany). Both recombinant proteins Tc964 and Lm964 were dialyzed against PBS [72].

### 4.7. Prediction of Linear B-cell Epitopes and Peptide Synthesis

Two synthetic peptides, TcNV and TcKP, derived from the Tc964 protein, were identified by the following bioinformatics tools as potential B epitopes: SVMTrip (version 1, Nebraska University, Lincoln, NE, USA; http://sysbio.unl.edu/) [73], Epitopia (http://epitopia.tau.ac.il/) [74], Bcepred (http://crdd.osdd.net/) [75], Bepipred-2.0 (http://www.cbs.dtu.dk/) [76] and ABC preds (version 3, Indraprastha Institute of Information Technology, New Delhi, India; http://crdd.osdd.net/) [77]. The peptides were synthesized and purified at the Curauma Nucleus Biotechnology Center of Pontificia Universidad Católica of Valparaíso, Chile, as described by Guzmán et al. [78].

### 4.8. Sera Samples and Ethical Statement

A total of 23 sera from patients with a confirmed diagnosis of ACL and CCL, all of them with a positive Leishmanin Skin Test, were provided by the research group in immunotoxicology, Universidad Nacional de Colombia, Bogotá, which were grouped as follows: 12 sera from patients with ACL, 11 from CCL patients; 6 sera from inhabitants of non-endemic areas and with a LST negative were used as negative controls. Meanwhile, 63 sera from patients diagnosed with CCD and with different stages of severity, classified as asymptomatic (Groups A and B) and symptomatic (Groups C and D) according to the American College of Cardiology and American Heart Association Staging [79,80] were obtained from the research group, in infectious diseases, Pontificia Universidad Javeriana, Bogotá, which were grouped as follows: 25 sera belonged to A group (normal electrocardiogram findings, normal heart size, normal left ventricular ejection fraction, New York Heart Association class I), 11 to B group (abnormal electrocardiogram findings, normal heart size, normal left ventricular ejection fraction, New York Heart Association class I), 18 to C group (abnormal electrocardiogram findings, increased heart size, decreased left ventricular ejection fraction, New York Heart Association class II-III) and 9 to D group (abnormal electrocardiogram findings, increased heart size, decreased left ventricular ejection fraction, New York Heart Association class IV) [81]. In total, 6 samples of sera from individuals that were inhabitants of non-endemic areas with diagnoses negative for CD were used as negative controls. All patients signed an informed consent form and their associated data were anonymized. This study was conducted in accordance with the Declaration of Helsinki, and the protocol was approved by the Ethics Committee from the Facultad de Ciencias of Pontificia Universidad Javeriana, Fundación Clínica Abood Shaio, Instituto Nacional de Salud and Hospital Universitario San Ignacio (Project code 120349326159) and from the Facultad de Ciencias, of Universidad Nacional de Colombia (Project code 37153).

### 4.9. ELISA Assay Protocol

The protocol initially described by Barral-Netto et al. [82] and adapted by Souza et al. [65] was used to develop the ELISA technique. In general, 96-well plates were coated with 100 µL/ well of soluble antigen (10 µg/mL), recombinant proteins (rTc964, rLm964 and Trigger Factor) (1 µg/mL), or the peptides (25 µg/mL) in carbonate buffer (0.45 M NaHCO_3_, 0.02 M Na_2_CO_3_, pH 9.6) overnight at 4 °C. After four washes with PBS 1×−0.05% Tween, the plates were blocked for two hours at room temperature with PBS 1×−0.05% Tween 20-1% bovine serum albumin (Eurobio, Les Ulis, France). After four washes with PBS 1×−0.05% Tween, sera were diluted 1:100 with PBS1×−0.05% Tween and 100 µL/well were incubated for one hour at 37 °C under shaking. The sera for the ELISA peptide assay were incubated for two hours. After five washes with PBS 1×−0.05% Tween, the wells were incubated with 100 µL/well of alkaline-phosphatase-conjugated anti-human IgG (Sigma, St. Louis, MO, USA) at a 1:2500 dilution in PBS1×−0.05%, 0.25% bovine serum albumin (Eurobio, Les Ulis, France) for one hour at 37 °C. Following the five washes with PBS 1×−0.05% Tween, the binding of the alkaline-phosphatase conjugate was determined by adding 100 µL/well of a chromogenic solution of p-nitrophenyl phosphate (pNPP) (Sigma, St. Louis, MO, USA) in sodium carbonate buffer pH 9.6 with 1 mg/mL of MgCl_2_ incubated by 30 min at room temperature. The reaction was stopped with 50 µL/well of 3M NaOH. Finally, the readings were done at 405 nm wavelength. The serological experiments were repeated twice for soluble antigen and recombinant proteins and three times for the ELISA peptide assay observing similar results.

### 4.10. Statistical and Data Analysis

The results from the ELISA test were expressed as RI, calculated as the optical density of each sample divided by the cut-off point (average of healthy controls + 2 standard deviations). RI values above 1 were taken as positive samples. Statistics were applied according to the particular distribution of the analyzed datasets. The significance between the two groups was calculated using the Mann–Whitney U test. Differences among subject groups were evaluated using Kruskal–Wallis and Dunn’s posttest for multiple comparisons. The Mann–Whitney U test was two-tailed, and the statistical significance was established with a value *p* < 0.05. The software GraphPad Prism 8 version 8.2.0 for Mac OS X (GraphPad, San Diego, CA, USA; www.graphpad.com), was used for the statistical analysis.

## 5. Conclusions

The results obtained allowed the characterization of the Tc964 protein of *T. cruzi*, establishing that this is a specific protein of Kinetoplastids that shows a nucleocytoplasmic location and has homology with the GTPase family proteins. It is important to highlight its potential use as an antigen for serodiagnosis and as an early biomarker of disease progression. In this scenario, the functional characterization of the Tc964 protein and its validation as a diagnosis or prognosis candidate are going to be the next steps.

## Figures and Tables

**Figure 1 ijms-21-02432-f001:**
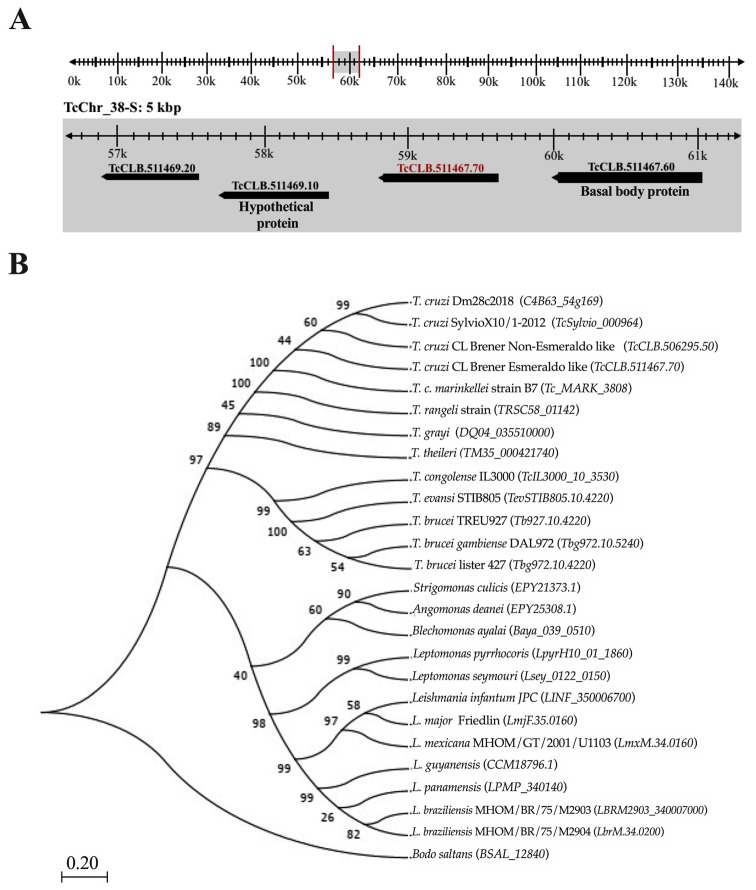
Localization of the gene coding for Tc964 protein in the *T. cruzi* genome and evolutionary relationship of this protein in different species of trypanosomatids. (**A**) Location of *TcCLB.511467.70* gene (red label) on the 38-S chromosome of *T. cruzi* (TcChr_38S), from curated reference strain CL Brener Esmeraldo like, with the coordinates 58,780 to 59,715k. (**B**) Maximum Likelihood phylogenetic tree proposed to the evolutionary history of the Tc964 protein among taxa of the Trypanosomatidae family; the tree was obtained with the Jones-Taylor-Thornton model using 1000 replicates of Bootstrap and a site coverage cutoff of 75% with the MEGA 7 program [34]. The selected outgroup was the *B. saltans* species. Numbers at the branches indicate maximum likelihood bootstrap support.

**Figure 2 ijms-21-02432-f002:**
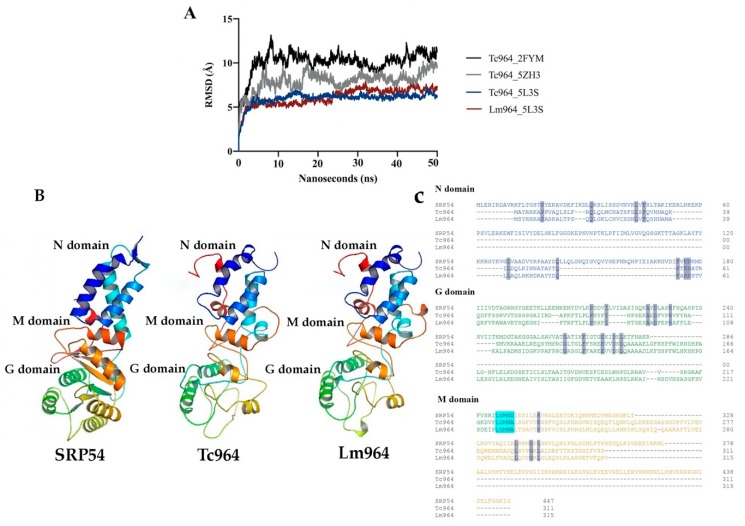
Molecular dynamics and structures of Tc964 and Lm964 proteins. (**A**) Root Mean Square Deviation values obtained during the 50 nanoseconds (2500 frames) of unrestricted molecular dynamics trajectories of the obtained models. Black line: Tc964_2FYM (template PDB code:2FYM) [38]. Gray line: Tc964_5ZH3 (template PDB code:5ZH3) [39]. Blue and red lines: Tc964_5L3S, Lm964_5L3S (template PDB code:5L3S) [37]. (**B**) Structural models of the 964 proteins from *T. cruzi* and *L. major.* Left: 3D Structure of the template SRP54 (PDB code:5L3S). Center: 3D structure model for Tc964 protein. Right: 3D structure model for Lm964 protein. N, G and M domains are located on the structures. 3D figures were generated using PyMOL Molecular Graphics System, (version 2.0, Schrödinger LLC, Portland, OR, USA) [43]. (**C**) Alignment between the sequence of SRP54 template protein and the 964 proteins of *T. cruzi* and *L. major using* the Omega Clustal program [44]. The domains N, G, and M are shown with the representative colors of the 3D model. Identical amino acids are highlighted in gray and the motif LGMGD is highlighted in alcian blue.

**Figure 3 ijms-21-02432-f003:**
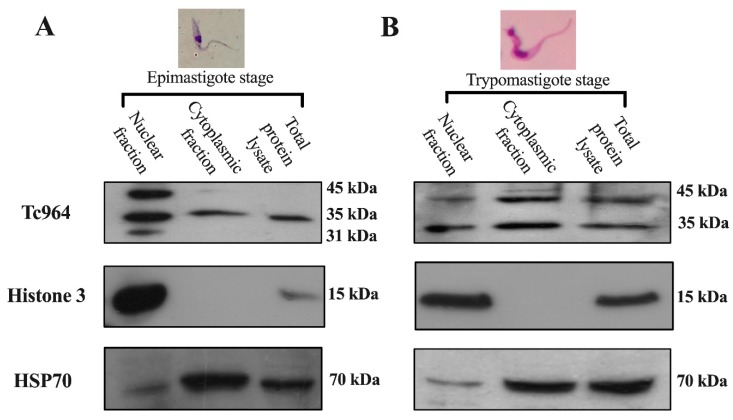
Subcellular localization of the Tc964 protein. (**A**) Detection of the Tc964 protein (35-kDa) in epimastigotes from the Y strain (DTU II) using an anti-Tc964 as primary antibody. (**B**) Tc964 detection on trypomastigotes strain Y (DTU II), with the anti-Tc964 antibody. Antibodies against Histone 3 (nuclei) and Heat-Shock Protein 70 (cytoplasm) were used as controls of subcellular fractionation.

**Figure 4 ijms-21-02432-f004:**
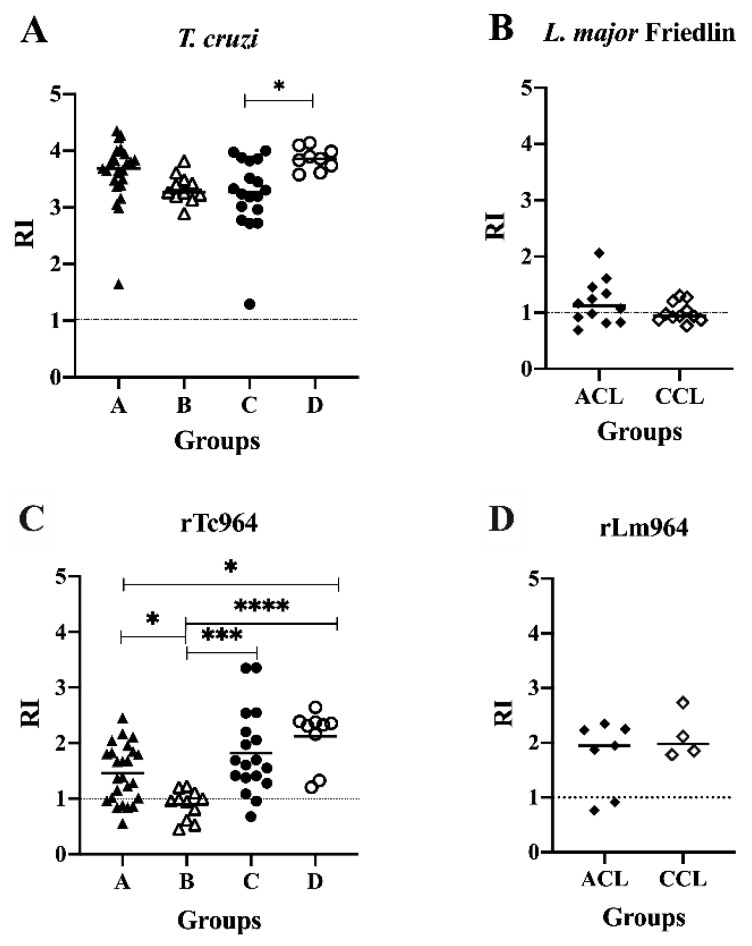
Antigenicity of the Tc964 and Lm964 recombinant proteins. (**A**)ELISA test, using a protein lysate of *T. cruzi* Y (DTU II) epimastigotes and sera from patients at different stages of CCD severity: Asymptomatic (A, *n* = 25; B, *n* = 11), symptomatic (C, *n* = 18; D, *n* = 9). (**B**) ELISA test, using a protein lysate of *L. major* promastigotes and sera from ACL (*n* = 12) and CCL (*n* = 11) patients. (**C**) ELISA test, using the rTc964 protein as antigen and sera from CCD patients at different stages of disease. (**D**) ELISA test, using rLm964 protein and sera from ACL (*n* = 7) and CCL (*n* = 4) patients. Mean values are indicated by horizontal lines. The RI corresponds to the average derived from two (2) replicas. The *p* values were calculated using the Mann-Whitney U test (ns *p* > 0.05) for (**B**) and (**D**) and one-way ANOVA nonparametric Kruskal–Wallis test with Dunn’s posttest (* *p* < 0.05, *** *p* < 0.001, **** *p* < 0.0001) for (**A**)and (**C**).

**Figure 5 ijms-21-02432-f005:**
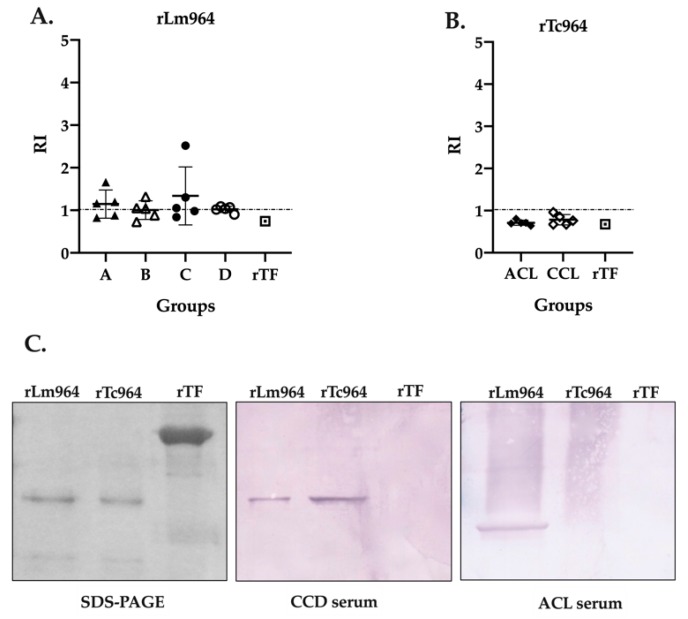
Cross-reactivity analysis of the Tc964 and Lm964 recombinant proteins. (**A**) ELISA test, using the rLm964 as antigen and sera of patients at different stages of CCD (**B**) ELISA test, using the rTc964 as antigen and sera of ACL and CCL patients. The recombinant protein Trigger Factor was used as negative control. Horizontal lines represent the mean ± SD of the values for each sera group. (**C**) Analysis of the antigenicity of the recombinant proteins rTc964 (37 kDa) and rLm964 (38 kDa) by Western blot. Left: 11% SDS-PAGE gel with the rTc964, rLm964 and recombinant protein Trigger Factor. Center: Western blot using the serum of a CCD patient with severity stage C (sample QX90). Right. Western blot using the serum of an ACL patient (sample 31A). In each assay, recombinant protein Trigger Factor was used as a negative control.

**Figure 6 ijms-21-02432-f006:**
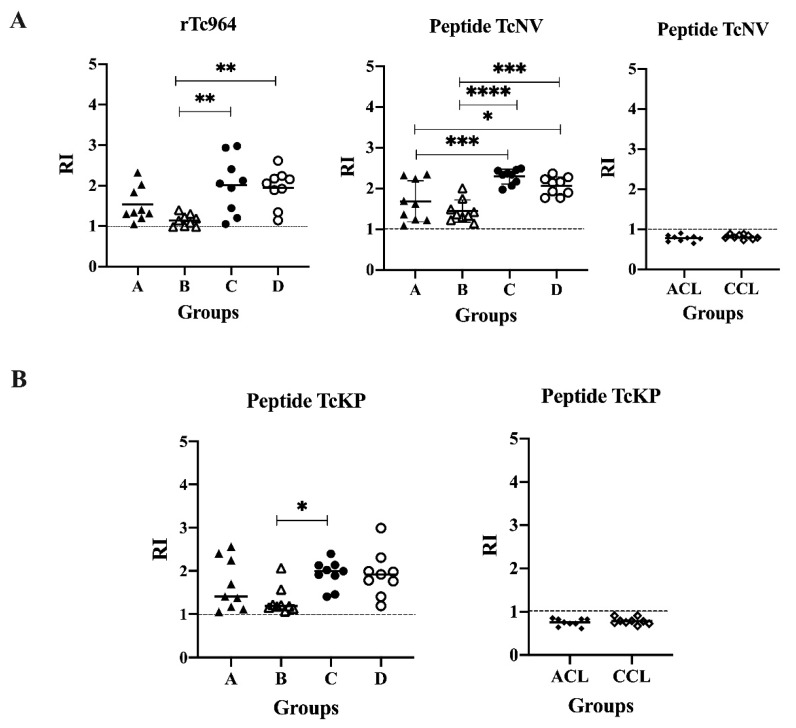
Antigenicity of the TcNV and TcKP peptides. (**A**) ELISA test, using the TcNV peptide and sera from patients at different stages of CCD and sera of ACL and CCL patients. The rTc964 protein was used as positive control. (**B**) ELISA test, using TcKP as antigen and as primary antibody sera from patients with CCD with different stages and sera of ACL and CCL patients. The RI values shown in the figures correspond to the average of three (3) replicas. The *p* values were calculated using one-way ANOVA nonparametric Kruskal–Wallis (ns *p* > 0.05, * *p* < 0.05, ** *p* < 0.01) for the rTc964, TcKP peptide vs CCD sera; one-way ANOVA Fisher’s LSD test (ns *p* > 0.05, * *p* < 0.05, *** *p* < 0.001, **** *p* < 0.0001) for the TcNV peptide vs CCD sera and the significance between two groups was calculated using the Mann-Whitney U test for both peptides vs ACL, CCL sera.

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
