# Peer review of "Molecular Characterization of Tc964, A Novel Antigenic Protein from Trypanosoma cruzi"

_ijms, 2020, doi:10.3390/ijms21072432_

Round 1
Reviewer 1 Report
General comment
This is an important study which was performed rigorously and using technically-sound approaches with regard to molecular and biochemical aspects. On the other hand, the preliminary validation of serological reactivity, including cross-reactivity, suffers from the lack of knowledge – or the non-availability(?) - of the broad spectrum of diseases represented by the main counterpart of Chagas disease, i.e. the human leishmaniases.
A suggestion could be to restrict this article to the well-conducted molecular characterization of the novel T. cruzi protein, and to preliminary results of Tc964 reactivity with sera from different groups of chronic chagasic patients. The crucial study on clinical serospecificity should be postponed to allow adequate inclusion of other diseases besides a spectrum of leishmaniasis conditions.
Specific comments
L30-34: this is a common clinical laboratory occurrence in homologous combinations of serum/parasite antigen when patients with localized cutaneous leishmaniasis are considered. For example, at least half of patients with active L.major lesions are seronegative (WHO 2010. Control of the leishmaniases). So why negligible/absent concentrations of antileishmanial antibodies should react with T.cruzi antigens? See more comments below
L74: “serodiagnosis of CD and leishmaniasis”: which clinical form of leishmaniasis?
L88: comment as above
L359: Results and Discussion of data obtained without appropriate Methodology could not be meaningful, therefore I am jumping from Introduction to M&M
L388, 4.3.2 paragraph: No justification is given here for the choice of L. major as the representative Leishmania species in this study on T. cruzi. Among the 21 species currently reported to infect humans (Akhoundi et al 2017. Mol Aspects Med) the probability of co-presence of L.major with T.cruzi in endemic settings or migratory patterns is near to zero. The most important species with clinical relevance for specific diagnosis is Leishmania infantum (see more comments below)
L490-494, sera samples: human leishmaniases includes several clinical forms, differing in the humoral response to the specific leishmanial agent and hence in the amount of specific antibodies detected by serodiagnosis.
Please note that positivity to LST is not indicative of antileishmanial antibodies presence and amount in a patient’s serum.
Among the clinical forms, localized CL like the one considered in this study (commonly caused by L.mexicana in the New World or L. major in the Old World) is the least “antibody productive” condition with around 50% seroreactive patients with active lesions, less when they are cured. The spectrum changes with more invasive Leishmania, such as Viannia parasites in the New World or L. aethiopica in the Old World, up to the opposite situation characteristic of visceral leishmaniasis. In this condition, patients affected by L.donovani complex (including L.infantum) are 100% reactive with homologous antigens and also at high rate with T. cruzi antigens.
These considerations imply that this study is severely lacking of clinical samples from different leishmaniasis conditions (and other infectious and non-infectious diseases).
L221: Reactivity Index in ELISA can be appropriate for preliminary comparative evaluation of reactivity and cross-reactivity of antigens, but it has no value for a clinical application in serodiagnosis, where a cut-off must be calculated
Author Response
Cover letter+ ResponsesToReviewer1.

Reviewer 2 Report
The manuscript by Ruiz-Márvez and colleagues describes a study on molecular characterization of Tc964, a novel antigenic protein from Trypanosoma cruzi. Overall the manuscript deals with an interesting topic that has not been studied much before. The manuscript’s English level is adequate. Data of the study indicate that Tc964 is indeed antigenic, and B-cell epitopes are mapped by the use of synthetic peptides. Tc964 was recognized by sera from Chronic CD (CCD) patients at different stages of disease severity.
Therefore, Tc964 would be adequate for CD diagnosis in areas where both infections (CD and leishmaniasis) coexist. The technical content of the paper is sufficient quality for publication.
Author Response
Cover Letter+ResponsesToReviewer2

Reviewer 3 Report
Manuscript Number: ijms-724564- Ruiz-Márvez et al. 26th February 2020
Major revision
The authors present data of a very interesting topic, the molecular characterization of a specific protein of Trypanosoma cruzi. A general suggestion is the avoidance of unusual abbreviations. The authors should control how often they used the respective abbreviation. If they used it less than 6 times, it should not be used. The readers don´t have to think about the sense of the respective abbreviation. However, often used abbreviations should be explained only once!!! Another general suggestion is the space between two citations. According to the Instructions this should be deleted throughout.
The Discussion is very focussed. The authors should include a short chapter of antigens used for diagnosis and commercially available diagnosis kits.
In the following, I include comments which refer to the respective line.
22, 26, 29, 86, 87, 102, 136, 148, 152, 161, 165, 276, 305, 308 and 352: The authors should re-write these sentences by deleting the verb before “ that” and deleting “that”. They should simply add “According” or “In” at the beginning of the sentences. Sometimes they can delete the first part of these sentences. Thereby, they avoid the extensive use of identical verbs and shorten the text.
55: The authors should replace “can be observed by microscopy” by “can often be observed by light microscopy”.
58: The authors should replace “after 10 to 30 years of” by “10 to 30 years after”.
93-97: The authors should place these sentences into the Introduction. Usually, citations should be avoided in the Results.
111-112 Fig. 1: The authors should write all names of species in italics. They should replace “Tc marinkellei” by “T. c. marinkellei”. In addition, they should replace “L. infantum” by “Leishmania infantum” and abbreviate the genus name Leishmania in the following lines.
302: The authors should include the aim to use the protein for diagnosis.
320: The authors should replace “shared ed” by “shared”.
362, 428: The authors should replace “964” by “Tc964”.
377-477: The authors should delete “Inc.,” and write the names of chemicals in lower case throughout.
390: The authors should include the relevant data of cultivation. What had been the periods of passages? This excludes or enables the development of metacyclic trypomastigotes!!!!!
390-399: The authors should place this part under the subheading “4.3.1 Cultivation of parasites” at line 376.
396: The authors should write the “2” in carbondioxide in lower case.
450: The authors should replace “contains” by “contain”.
453, 466: The authors should replace “showed” by “shown”.
469: The authors should replace “nzytech genes & enzymes Lisbon” by “nzytech genes & enzymes, Lisbon”.
514: The authors should include the authors before the citations.
524: The authors should replace “revealed” by a commonly used term.
527: The authors should replace “NaOH 3M” by “3M NaOH“.
622: The authors should correct the mistake.
776: The authors should correct the citation.
570-778: The references need a correction according to the Instructions. I will only mention general mistakes:
- “[Internet]” should be deleted throughout.
- English titles should be written in lower case throughout.
- After a double point, English titles should be written in lower case throughout.
- Between pages, the authors should replace the short hyphen by the long dash.
- The authors should read a recent publication of this journal whether or not abbreviated journals include a point after the abbreviation or not. This should be corrected throughout. This is also necessary for formatting “year, volume and pages” with or without a space between the numbers.
- The authors should abbreviate journals throughout.
- The authors should write the year in bold.
- The authors should write only genus and species names in italics (wrong: e.g. 637).
- The authors should write subspecies names in italics (wrong: e.g. 656).
10: The authors should include in numberings of pages all numbers (wrong: e.g. 681).
11: The authors should correct citations of book chapters according to the Instructions throughout.
Author Response
Cover letter+ResponsesToReviewer3.

Round 2
Reviewer 1 Report
Line 298: according to taxonomy rules, Family (like Trypanosomatidae) should not be written in Italic (only Genus and Species)
Author Response
Specific comments
-Line 298: according to taxonomy rules, Family (like Trypanosomatidae) should not be written in Italic (only Genus and Species).
Answer: Thanks! Following the reviewer's suggestion, the word Trypanosomatidae, initially written in italics was corrected throughout the text (lines 23,116,296,359,364).
Reviewer 3 Report
Manuscript Number: ijms-724564- Ruiz-Márvez et al. 10th March 2020
Minor revision
The revision has strongly improved the manuscript. My general suggestion, the avoidance of unusual abbreviations, didn´t include abbreviations generally used in molecular biology. Therefore, “messenger RNA” should be replaced throughout by “mRNA”. My second general suggestion, the deletion of the space between two citations, hasn´t been corrected throughout (e.g. not in line 44).
In the following, I include comments which refer to the respective line.
83-84: The authors should replace “in triatomine insects, LYT1 modulates” by “LYT1 is invlolved”.
84: The authors should replace “stage” by “stage, usually ocurring in the triatomine insect vectors”. [The changes in lines 83 and 84 avoid an over-interpretation of data obtained only by using cultured parasites.]
222, 226: The authors should replace “antigen” by “antigens”.
393: The authors should replace “dulbecco” by “Dulbecco”.
578-796: The references still need a correction according to the Instructions. The following previous suggestions had not been considered throughout.:
- Between pages, the authors should replace the short hyphen by the long dash (lines 580, 712).
- The authors should read a recent publication of this journal whether or not abbreviated journals include a point after the abbreviation or not. This should be corrected throughout. Many abbreviations aren´t followed by a point, especially the last abbreviation.
603: The authors should replace “ras” by “Ras”.
638: The authors should replace the short hyphen by the long dash.
651: This hasn´t been cited in the text.
694: The authors should replace “american” by “American”.
706, 770: The authors should replace “ONE” by “One”.
733: The authors should replace “S.G” by “S.G.”.
743: The authors should replace “Kinetoplastid” by “kinetoplastid”.
748: The authors should replace “Genome” by “genome”.
754: The authors should delete both quotation marks.
781: The authors should replace “butcher” by “Butcher”.
Author Response
General comment
The revision has strongly improved the manuscript. My general suggestion, the avoidance of unusual abbreviations, didn´t include abbreviations generally used in molecular biology. Therefore, “messenger RNA” should be replaced throughout by “mRNA”. My second general suggestion, the deletion of the space between two citations, hasn´t been corrected throughout (e.g. not in line 44).
Answer: Thank you very much for the detailed reviewing of the manuscript writing. We are really grateful for your efforts, and now we have double-checked your indications with the aim of eliminating all the errors.
Following the reviewer´s suggestion, “messenger RNA” was replaced by “mRNA”, throughout the text (lines 21,81,86,307), and the space between two references in the main text was also corrected throughout the text (lines 44,46,305,325).
Specific comments
In the following, I include comments which refer to the respective line.
83-84: The authors should replace “in triatomine insects, LYT1 modulates” by “LYT1 is involved”.
Answer: as suggested, the phrase was replaced by “LYT1 is involved in the transition from the epimastigote to the metacyclic trypomastigote” (lines 83-84). 84: The authors should replace “stage” by “stage, usually occurring in the triatomine insect vectors”. [The changes in lines 83 and 84 avoid an over-interpretation of data obtained only by using cultured parasites].
Answer: following the reviewer´s suggestion, the word “stage” was replaced on the text as follows: “In addition, LYT1 is involved in the transition from the epimastigote to the metacyclic trypomastigote stage, usually occurring in the triatomine insect vectors” [20-22] (lines 83-85).